# Controlling the thickness of the atherosclerotic plaque by statin medication

Dorota Formanowicz[1][⦾]*, Jacek B. Krawczyk[2][⦾]*

**1** Department of Clinical Biochemistry and Laboratory Medicine, Poznan University of Medical Sciences, Poznan, Poland, **2** School of Mathematics and Statistics, The University of Sydney, Sydney, NSW, Australia

⦾ These authors contributed equally to this work.
* doforman@ump.edu.pl (DF); jacek.krawczyk@sydney.edu.au (JBK)

**Data Availability Statement:** All relevant data are previously published and referenced within the manuscript.

**Funding:** This research has been partially supported by the National Science Centre (Poland)

## Abstract

Atherosclerosis, a chronic inflammatory disorder of the arterial wall, is a complex process whose dynamics are affected by multiple factors. The disease control consists of restraining it by administering statins. Slowing down or halting the plaque growth depends on the patient age at which the statin treatment begins and on the thickness of the intima-media (IMT) at that time. In this paper, we propose a mathematical model to estimate the sets of atherosclerosis states, from which the use of statins can restrain the disease. Our model is control-theoretic, and the estimated sets are the *viability kernels*, in the parlance of viability theory. To our best knowledge, this way of modelling the atherosclerosis progression is original. We compute two viability kernels, each for a different statin-treatment dose. Each kernel is composed of the vector [*age*, IMT] from which the disease can be restrained. By extension, the disease can't be restrained from the kernel complements, this being mainly because of the disease and patient-age advancement. The kernels visualise tradeoffs between early and late treatments, which helps the clinician to decide when to start the statin treatment and which statin dose may be sufficient.

## Introduction

The aim of this paper is to use a control-theoretic aggregate model of the progression of atherosclerotic plaque to assess viability of the process of restraining it. Atherosclerosis is a multifactorial and multistep disorder driven by chronic inflammation within the arterial wall. Therapy consists of treating patients with statins to achieve stabilisation of the plaque at a survival level.

However, the treatment may not improve, or stabilise, the patient's wellbeing if the plaque is thick or if the patient is at an advanced age. On the other hand, the treatment may be nonessential for young patients, or when the plaque is thin. Our aim is to estimate the disease states [*age*, IMT] (where IMT is the obvious abbreviation for *intima-media thickness*), from which the statin treatment is advantageous to patients' wellbeing. We call these sets the *viability kernels*.

Grant No. 2012/07/B/ST6/01537 (DF) and by the
statutory funds of Poznan University of Medical
Sciences (DF).

**Competing interests:** The authors have declared
that no competing interests exist.

There are several mathematical models that study the blood flow dynamics in arteries, oxidative disturbances, cholesterol formation, etc., see e.g., [1–8]. Those models help understand the dynamics of plaque formation at a *micro* level. In contrast, [9] propose a *macro* model. I.e., their model concentrates on patient holistic wellbeing related to atherosclerosis. Furthermore, a single feature that makes their model significantly different from the above micro models is that it includes a *control variable*, in the *control-theoretic* sense. That control variable is the amount of statins administered to a patient. It is this feature that enables us to compute the *viability kernels*, see e.g., [10] or [11], which—in this paper—are selections of atherosclerotic conditions, from which the disease can be relieved or stopped. Respectively, the kernel complements consist of the states, from which the model predicts that the plaque stabilisation is impossible.

Here is a brief outline of what this paper contains. We first provide a brief on the pathophysiology of atherogenesis. In the subsequent section, we present atherosclerosis as a dynamic process and propose a mathematical model that captures the basic features of atherosclerosis progression and the role of statins in slowing down this progression. We then show and discuss a few time profiles of IMT obtained from our model. Next, we compute and display two viability kernels that contain the atherosclerosis states, from which one can slow down the disease process. Concluding remarks and methodological comments follow. In an appendix, we review a number of studies on possible effects of cholesterol on atherosclerosis.

## The essence of atherogenesis

The pathophysiology of atherogenesis is a complex phenomenon. It is widely acknowledged that it can be decomposed into the following key components, or stages:

1. endothelial cell injury followed by endothelial dysfunction, see [12];

2. lipoproteins deposition and their oxidation, see [13];

3. inflammation of the arterial wall, see [14];

4. fibrous cap formation, see [15].

We contend that our aggregate model captures the process' consequences.

As it will become evident in the next section, our mathematical model for atherosclerosis progression is silent about cholesterol. Notwithstanding cholesterol's widely recognised role in explaining the pathogenesis of the atherosclerotic plaque we refer to the *oxidative response to inflammation*, see e.g., [13, 16], which consists of inflammation of the arterial wall, and may be is considered the main cause of the disease. Indeed, inflammation is crucial for instability of the plaque and clinical consequences of its rupture. This enables us to use statins that—in a real patient—*suppress* inflammation and prevent or delay the fibrous cap formation, and—in our model—*control* the fibrous cap thickness. However, we acknowledge the frequent opinion that cholesterol may play an important role in promoting atherosclerosis. We therefore provide justification for our line of research. Below, we speak about inflammation and how it helps understand the plaque growth. In the appendix we present results of several studies on a possible impact—or its lack—of cholesterol on atherosclerosis.

### Inflammation

The response-to-injury theory, see [17], can be seen as a base for our model. This theory proposes that atherosclerosis is an inflammatory response to the injury to the endothelial cells, which line all blood vessels, that becomes the driving force for the entire process of atherogenesis, see [18].

Dysfunctional endothelium, affected by monocyte-macrophage infiltration, is an essential prerequisite for the formation of a fibrous cap, whose integrity is crucial for atherosclerotic plaque clinical consequences. Many hypotheses try to explain the mechanisms underlying plaque rupture. Each of them seems to have advantages, and each deems important a different component of an atherosclerotic plaque, i.e.,:

a. increased mechanical stress leading to fatigue of the extracellular matrix (the fibrous element of the cap);

b. modified lipoproteins promoting cell death and matrix destruction;

c. change in macrophage phenotype (M2 anti-inflammatory and M1 pro-inflammatory) as a response to microenvironment signals inside the plaque, studied widely in [19, 20];

d. vascular smooth muscle cell aberrant phenotypic transitions promoting vascular calcification;

e. loss of endothelial function leading to increased leukocyte infiltration and reduced smooth muscle cell chemoattractants;

f. the penetration of T- lymphocytes and mast cells adversely affecting the balance of cap formation and disintegration.

Differences in the contribution of individual factors to the formation of atherosclerotic plaque may result from various personal conditions, both genetic and epigenetic. Therefore, each of the above components will have a different weight in explaining a patient's condition.

We notice that the JUPITER trial (see [21, 22]) confirmed that men and women with elevated hsCRP (the high-sensitivity [hs] assays of C-reactive protein [CRP]) and low LDL-C, are at a substantial vascular risk, and demonstrated that the statin therapy can cut that risk by half [22]. In [23, 24], beyond lipid lowering, statins were found to have anti-inflammatory properties, as evidenced by the diminished hsCRP scores. Based on the history of 3745 patients with acute coronary syndromes receiving statins (80 mg of atorvastatin or 40 mg of pravastatin per day), [25], evaluated (in PROVEIT-TIMI22 trial) the relationships between the LDL-C and CRP, and the risk of recurrent acute myocardial infarction (AMI) or death from cardiovascular disease (CVD). The study has revealed that patients whose CRP levels were lowered due to statin therapy, have better clinical outcomes than those with higher CRP, regardless of the level of LDL cholesterol. Overall, the benefits of the statin therapy occur primarily in patients with elevated hsCRP, see [26]. Moreover, the JUPITER trial results strongly suggest that elevated hsCRP levels, rather than other factors, best correlate with the high cardiovascular event rates observed in the trial, despite low LDL cholesterol levels, see [22]. Further, post hoc analyses from lipid-lowering trials have revealed that individuals reaching dual low LDL-C and hsCRP targets had better outcomes compared with those who only reached an LDL-C target, see [27, 28]. In addition, in a cohort of 4 000 adults, aged 45-64 years, without known atherosclerotic CVD at baseline and followed for above 18 years, [29] have discovered that individuals with a higher degree of inflammation, determined by high hsCRP, had heightened risk of atherosclerotic CVD events and all-cause death, across various levels of atherogenic lipid measures.

[30] report that none of the lipids ratios have been found to be informative in prediction AMI. The cited authors in the study performed on 67 patients with AMI and 25 controls, show that the reduction in serum total cholesterol does not prevent the cardiovascular risk. They also show that systemic inflammation increases this risk.

Furthermore, [31] (and other studies, see below) report a data conflict concerning the exact nature of the relationship between atherogenic LDL-C and CVD, especially in the elderly.

Clearly, there is data conflict with regard to the precise nature of this relationship in elderly persons. In particular, [32–34] tell of this relation as U-shaped; [34, 35] report it is as J-shaped, and [36] find evidence that it is linear. Some studies, like [37, 38], cannot find any such association. In the review paper [35], a detailed analysis of the relationship between all-cause and cardiovascular mortality on the one hand, and LDL-C on the other is based on the results of studies of 30 cohorts including 68,094 elderly people. The authors point to statistical significance of inverse association between all-cause mortality and LDL-C in 14 cohorts. Interestingly for us, CVD mortality was highest in the lowest LDL-C quartile in two cohorts. (No association was found in 7 cohorts.) We, therefore, conclude that generalisations regarding the existence of a strong association between high LDL-C levels and CVD, as well as its clinical consequences, are not supported by the data.

### Evidence

We conclude there is evidence that normal hsCRP levels correspond to thinner or more stable plaque. As a consequence, we contend that inflammation and its treatment by statins can explain the progression and stabilisation of atherosclerosis.

There is also support for the view that the statin treatment decreases both: inflammation and cholesterol levels. Recently, [39–41] have found that the use of statins as the anti-inflammatory, anti-oxidant, and immuno-modulatory drugs, in addition to their known lipid-lowering effect, proved to be beneficial in the context of many diseases. In particular, these authors assert that statins exert a direct anti-atherosclerotic effect on the arterial wall, beyond their effects on plasma lipids. In our view, this is of great consequence for the use of statins in atherosclerosis—a chronic inflammatory disease. Light-heartedly, [42] says that hypercholesterolaemia and the inflammatory process are "partners in crime". We propose that our model and the conclusions drawn from it will be attractive to both.

## Atherosclerosis as a dynamic process

In this section we analyse atherosclerosis' dynamics. As in [9], we are interested in the effects of statin treatment on patients' IMT. Here, however, we propose that atherosclerosis is a two-dimensional dynamic process, rather than one-dimensional as in the above publication. The two dimensions of the model are: the patient current IMT level, and the long-term, perhaps terminal, level of the plaque thickness. It is this second dimension that can be impacted upon by a statin treatment. In the model, the long-term thickness and the current IMT interact with each other, which is what we contend takes place in real life.

### S-shaped growth

We refer to [9] for a more detailed description of the carotid plaque progression. Here, we highlight only its main features.

Taking into account the significance of the lumen of blood vessels in the circulation, and the role of plaque in its reduction, we will treat IMT as the main model variable, which—if large—identifies population of high risk of CV (even if the more traditional risk factors might suggest the opposite).

We contend that there is clinical evidence of an S-shape of the IMT time profile over a patient life span. Here we point to [43] who show that the plaque growth is fastest in the middle-age group and to [44] who report little or no growth of the plaque in centenarians.

Although there is little research into the children's and youths' growth of IMT, it is known, see [45], that the growth can begin in childhood as an accumulation of fatty streaks-lipid-engorged macrophages and T lymphocytes in the intima of the arteries. Fatty streaks may or

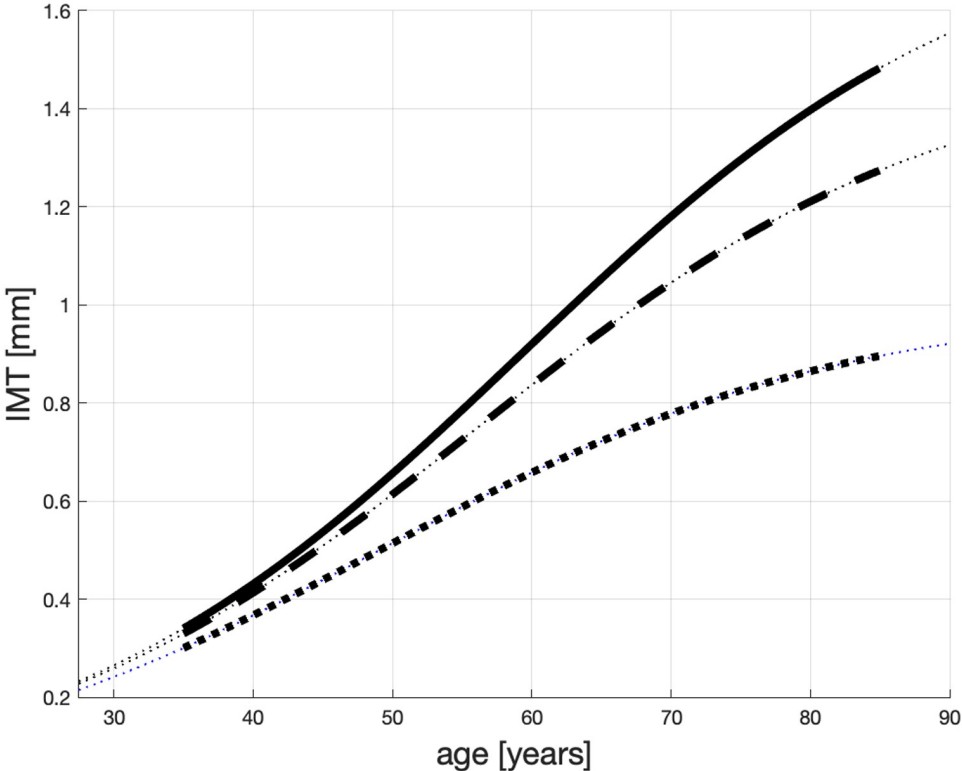

**Fig 1. Possible IMT time profiles for persons with different atherosclerosis condition.**

may not progress; they may also regress. Data on IMT changes at the age of 0-35 are difficult to collect and analyze because, in this group of people, there are no routine tests to assess the progression of atherosclerotic changes. Most often, such studies are carried out only at risk groups in children and young adults with a history of cardiovascular disease or such diseases in families. In brief, an analysis of the available IMT test results conducted on groups under 40 gave us quite vague and often mutually exclusive data, see [46–48].

Nonetheless, the IMT measurements in the first-time cardiology patients, whose mean age is 35-40 [yo], suggest that the plaque's growth is slow in young people. This, combined with the above literature on IMT growth in the middle-age and old patients, points to its fast growth between 45 and 80 [yo] and attenuation thereafter. This suggests strongly that a logistic differential equation can be an adequate mathematical model for the plaque's progression over the entire patient lifespan.

It is commonly accepted that the pace of IMT growth correlates well with traditional cardiovascular risk factors, like excess weight, dyslipidemia, and hypertension. By way of an example we show in Fig 1 three possible IMT time profiles for three groups of patients with different risk factors. The top line is for the highest risk group (e.g., on dialysis), the middle line is for a medium risk group and the bottom is for "healthy" patients.

Let $[0, \mathcal{T}]$ is a patient's life span, so $\mathcal{T}$ is their terminal age and $t \leq \mathcal{T}$—their current age. Let the process $x(t)$, $t \in [0, \mathcal{T}]$, $x(t) > 0$ represent the IMT buildup. Process $x(t)$ will be S-shaped if it is a solution to the following logistic differential equation:

$$\frac{dx}{dt} = ax\left(1 - \frac{x}{c}\right) \qquad x(0) = x_0. \tag{1}$$

Here $a > 0$, $c > 0$ and $x_0$ is small and positive. The parameter $a$ determines the speed of the plaque's buildup and can be gender specific (or race, or life-style, or BMI, etc.). Symbol $c$—called "carrying capacity" in population dynamics—is determined by the state of health of the patient and represents the terminal (maximum) size of the plaque when they pass away.

Examples of solutions to (1) are shown in Fig 1; see [9] for a more detailed description.

The lines shown in Fig 1 represent the life-span IMT progression in three representative patients. The dotted line, which converges to "carrying capacity" $c = 1$ mm—is for atherosclerosis non-sufferers, "healthy" patients; the dashed line, converging to $c = 1.5$ mm—for non-healthy but not necessarily "heavy" sufferers; the solid line, converging to $c = 1.8$ mm—is for "heavy" sufferers. The other parameters that generate these solutions to Eq (1) are $a = 0.06$ and $x_0 = 0.05$.

We can see in this figure that when a healthy patient is 40 yo—see the dotted line—his, or her, IMT reaches approximately 0.37 mm, and progresses slowly for the next 20 years (about 0.015 mm/yr). This patient might live above 100 years, like the healthy Japanese from [44], and die from other causes. For "heavy" sufferers e.g., patients on dialysis, see the solid line, the plaque grows approximately 0.025 mm/yr when they are between 40 and 60 [yo]. Along the dashed line, the plaque increases approximately 0.021 mm/yr when they are between 40 and 60 [yo] and their IMT may reach 1.5 mm, which is a typical terminal IMT value in non-heavy sufferers.

We notice that the speeds of the IMT growth shown in Fig 1, correspond to those reported in [44, 49, 50].

The one caveat concerns patients reported in [51, 52] who, in childhood, had higher IMT than shown in our Fig 1. Those patients suffered from familial hyperlipidemia, parents' history of premature AMI or accumulation of these and other early childhood or prenatal risk factors. In this paper we model the behaviour of patients who begin to have cardiovascular problems when they are about 45 [yo]. We contend they are the most representative group of patients with CVD.

## Statin-controlled terminal plaque size

The value to which IMT converges at the life's end (so, *terminal*) of a patient is $c$. It is naturally constrained: $c \in [c_H, \text{IMT}_{max}]$. Here $c_H > 0$ is an IMT value that is typical of *non-sufferers* ("healthy") when they are ninety or over, and $\text{IMT}_{max}$ is the plaque's thickness measured in the disease *sufferers* on dialysis at the end of their lives. Clinical evidence, see [53, 54], suggests that $c_H$ can be approximated by 1mm, and $\text{IMT}_{max} \approx 1.8$ mm. These are the values that we used to create Fig 1.

We contend that if the disease recedes or slows down, a patient's value of $c$ must have diminished. Of course, this effect cannot be measured and so, for us, $c$ is an *implicit* measure of patient wellbeing. We find this hypothesis convenient for explaining how statins may influence the pace of IMT growth in our *macro* model. In essence, we conjecture that

$$\text{statins} \Rightarrow \text{wellbeing} \Rightarrow \frac{dc}{dt} < 0 \Rightarrow \text{IMT converges to a lower } c. \tag{2}$$

We base this hypothesis on the comparisons between:
- the terminal IMTs of patients diagnosed as the heavy suffers and long-term recipients of statins, *and*
- the terminal IMTs of the patients who were not treated by statins (for a review see [55]).

In essence, we propound that growth of IMT depends primarily on $c \in [c_H, \text{IMT}_{max}]$, which is the parameter that "pulls up" each IMT curve in Fig 1. If a patient with a large $c$ receives

treatment or changes lifestyle, the parameter *c* can decrease and the IMT progression toward the plaque life-threatening levels will slow down. As a result, the patient may live longer.

In a two-dimensional model, which we propose in this paper, symbol *c* becomes a shorthand for a time-dependent process $c(t)$, $t \in [t_1, t_2]$, which constitutes the model's second dimension. In the model, this process depends on:

1. therapy by statins $s(t) > 0$; if there is no therapy, $s \equiv 0$ then $c = const$.

2. patient age *T* and IMT, at which the therapy begins.

   More precisely, we conjecture that

- statins administered to patient of age $T < \mathcal{T}$ modify process $c(t)$ in that $c(t)|_{s(t)>0, t\in[T,T]}$, i.e., $c(t)$ after the statin treatment started at age *T*, is *below* the original $c = const$.

- the older the patient and the larger their current IMT, the stronger the tendency of $c(t)$ to keep in line with the original $c = const$.

We offer the following differential equation to capture the evolution of $c(t)$, which fulfils the above conjectures:

$$\frac{dc}{dt} = \begin{cases} 0 & \text{for} & t < T \\[2em] \underbrace{-A(Bs(t)+C)c(T)\,e^{T-t}}_{(\bullet)} + \underbrace{\frac{DE\,(x(T)-x(40))}{D^2\,(T+\Delta-t)^2 + 1}}_{(\bullet\bullet)} & \text{for} & s(t) > 0, t \in [T, \mathcal{T}] \end{cases} \cdot (3)$$

We claim that this equation captures the process of the plaque growth; however, we do not suggest that this is a unique way of this process' representation.

Here is how Eq (3) can be interpreted.

The *first* line says that the IMT "pull-up" level remains unchanged if statins are not administered.

The *second* line describes what happens to $\frac{dc}{dt}$ that is to the pace of changes of the terminal plaque thickness, after a patient has begun statin treatment at age *T*. Here, *A*, *B*, *C*, *D*, *E* are calibrated constants. [56] has observed that the IMT diminishes in the first 3-4 years after the start of a statin therapy and then it grows; parameter $\Delta$ captures this process. This can be better understood by noticing that an integral of a fraction like (••) results in `arctan` and that the quick change from negative to positive values occurs at $t = T + \Delta$.

(•) This term is negative and causes $c(t)$ to diminish in $s(t) > 0$. It represents a "shock" of the patient's organism after statin therapy has begun. The depth of the shock depends on $c(T)$: the thicker the plaque at *T*, the stronger the organism reaction. If a statin dose remains unchanged in $[T, \mathcal{T}]$ then $c(T) = c$ is typical of the treated patient (e.g., heavy sufferer on dialysis). The statin effect dissipates exponentially with time *t*.

(••) This term is positive and contributes positively to the growth of $c(t)$. It takes into account the timing of statin therapy.

As observed above, this term attains maximum at patient age $t = T + \Delta$. Before this age, the sum (•) + (••) must be negative to diminish *c*. At $T + \Delta$ this sum will be positive, so *c* grows. After this time, the sum will tend to 0. This will determine a new level of *c* lower than the original $c(T)$, toward which IMT converges at the patient life-span.

Assuming that $\Delta = 3.5$ `years`, see [56], we show a simulated time profile of the IMT pace changes in Fig 2 top panel. The corresponding $c(t)$ is shown in bottom panel. Here, the statin treatment is of Atorvastatin 80 `mg/d` and begins when the patient is 60 years old. The values of calibrated parameters $A$, $B$, $C$, $D$, $E$ will be explained below. The resulting IMT life-span time profiles will be shown in Fig 3.

The new level of $c$ depends on the difference $x(T) - x(40)$ where $x(40)$ corresponds to an assumption—justified by medical practice—that statin therapy does not usually begin earlier than when the patient is 40 years old. The thicker the plaque $x(T)$ at the time the therapy starts, relative to the patient's plaque when they were 40, the higher the new value of $c$.

## Calibration

Our model is patient-group specific e.g., $x(40)$ for heavy sufferers will be different from mild sufferers', from healthy patients', etc. We allow for these observations in the model calibration.

The values for $a$, $x_0$ and the original $c$ are the same as in [9] i.e., $a = 0.06$, $x_0 = 0.05$; parameter $c = 1.678$mm, which is less than 1.8mm that is our model upper limit of the plaque, but it still characterises heavy sufferers. In the numerical simulations of our model, the values of $x(40) = 0.4278$mm and $x(T)$ also correspond to heavy sufferers. They are obtained from a solution to Eq (1) for $t = 40$ and $T$, i.e., the time when statin therapy starts.

The other model parameters are calibrated as follows: $A = 0.007407$, $B = 0.1135$, $C = 70.92$, $D = 24$, $E = .7281/\pi$ and $\Delta = 3.5$. They can be interpreted by looking at the system (1)–(3) to see how they affect the patient. E.g., $x_0$ is the value of IMT in infants; $A$—amplifies the effect of patient's wellbeing at $T$; $B$—amplifies the statin effect; $C$—is an improvement in patient wellbeing after $T$ when $s > 0$; $D$ regulates the speed of stabilisation of $c$ at the new level, as a result of statin treatment; etc.

## The model variables

In summary, our model variables are:

**IMT** *carotid intima-media thickness*. As found in many large studies, such as, for example, the Kuopio Ischemic Heart Disease Risk Factor Study (KIHD), the Atherosclerosis Risk in Communities (ARIC), the Cardiovascular Health Study (CHS), the Rotterdam Study, and the Malmö Diet and Cancer Study (MDCS), there *exists* a relationship between the carotid IMT value and the risk of cardiovascular events, see e.g., [57]. For an overview of the studies on this relationship see [58].

Atherosclerotic plaque deposition is a *dynamic* process where the existing thickness depends on the thickness from the previous stage. Due to these dynamics, atherosclerotic plaque deposits will be treated as a *state* variable in our control-theoretic model. We propose Eq (1) to capture the patient life-span plaque deposits profile.

In clinical conditions, *carotid intima-media thickness* (IMT) is measured using a high-resolution B-mode ultrasound method and treated as a proxy for both the plaque advancement and survival of the patient. (We notice that *plaque volume* (PV) has been also measured, see [59]. However, it is IMT which is an emergent risk marker of generalised atherosclerosis [60], which we use in this paper.)

**c** *terminal plaque size*. We hypothesise that plaque thickness is a function of patient wellbeing. The thickness converges, by the end of patient life, to a terminal plaque size charateristic of the patient type: heavy sufferer, non-heavy sufferer, healthy. The terminal plaque size is the second state variable in the model. The wellbeing can be impacted by statin treatment and

(a)

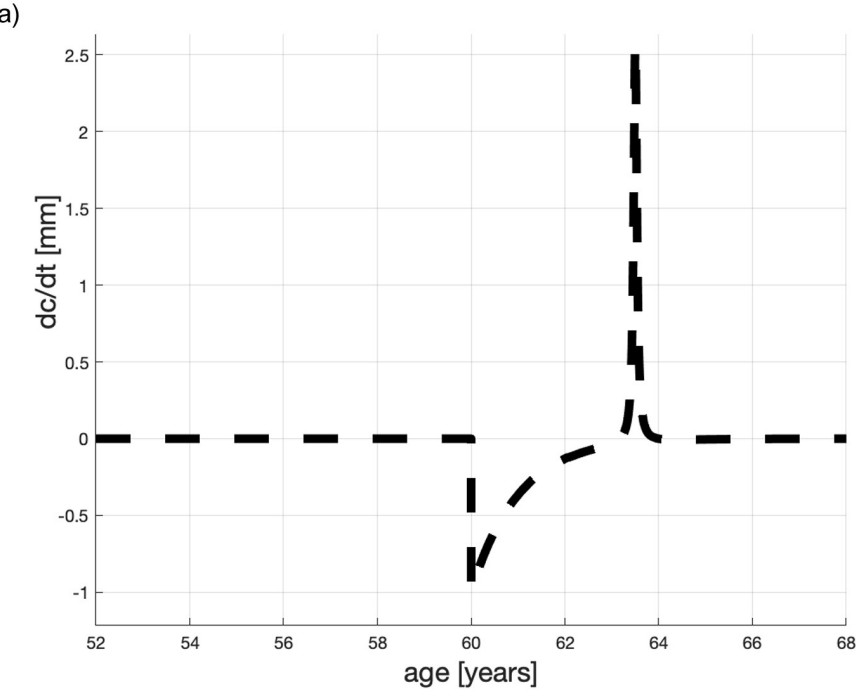

(b)

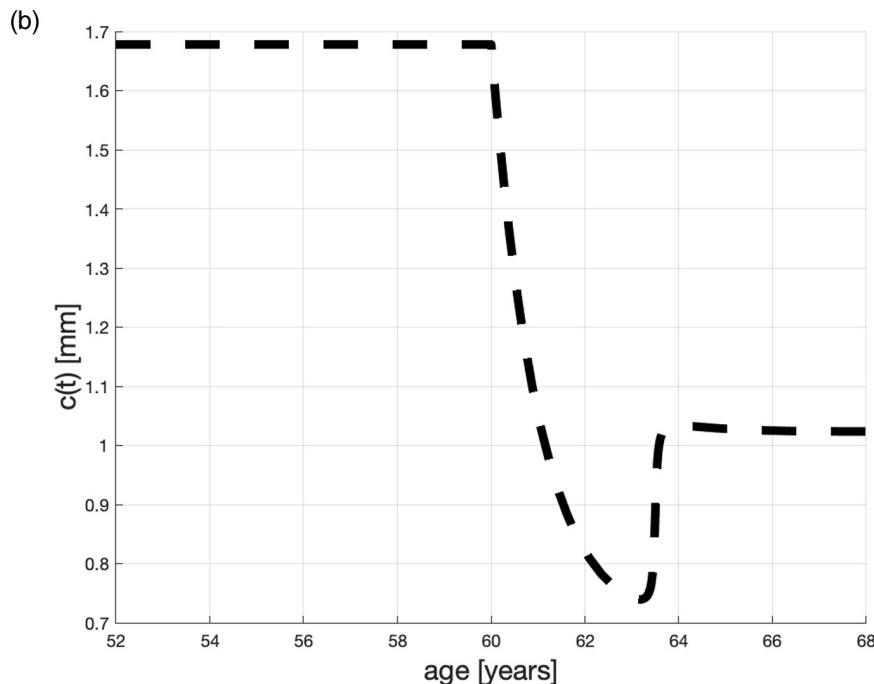

Fig 2. Calibrated time profiles of $\dfrac{dc}{dt}$ and $c(t)$ for a heavy sufferer treated from $T = 60$ years.

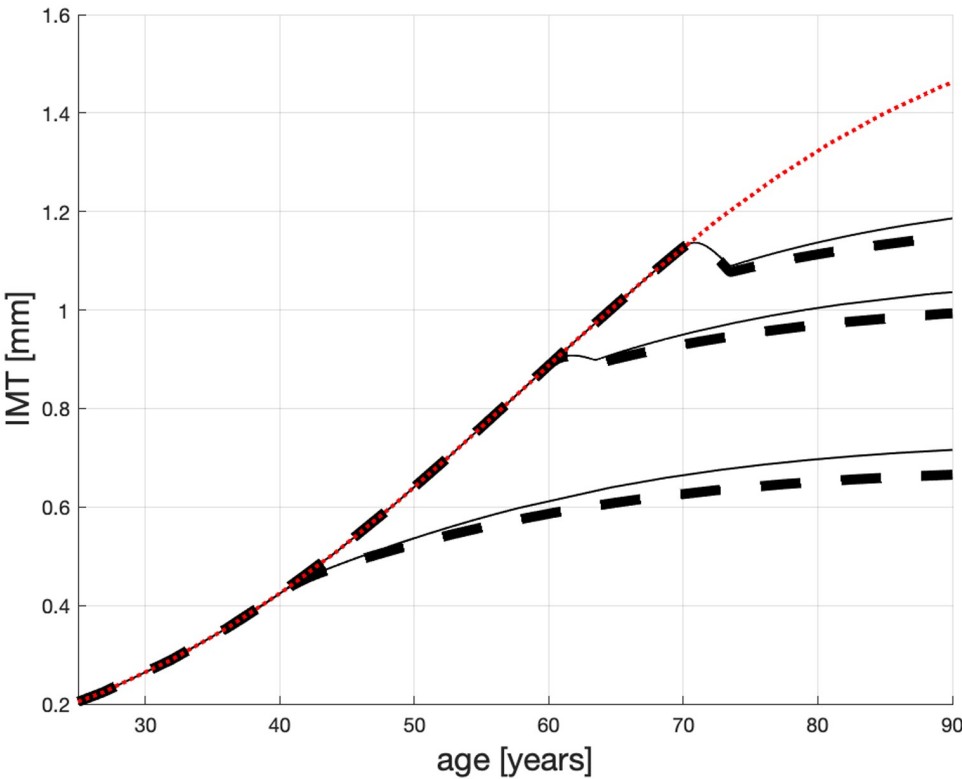

**Fig 3. Possible age profiles for male patients on maintenance dialysis treated with atorvastatin: 80 mg/d—dashed line; 40 mg/d—solid line.**

also by life-style changes. We have proposed Eq (3) to describe the stain impact on the terminal plaque size.

**Statins** Statins exhibit many pleiotropic activities, like anti-atherothrombotic, lipid-lowering and anti-inflammatory, see [61, 62]. One of the key non lipid-lowering effects of statins is the improvement of parameters related to the endothelial function. Notably, endothelial dysfunction is inimical to all atherosclerosis-underlying phenomena, thus to cardiovascular events. Much of the recent experimental and clinical data have shown that the anti-inflammatory effects of statins, even without significantly affecting the lipid levels, can actually improve the prognosis of patients with CVD. Post hoc subgroup analysis in the West of Scotland Coronary Prevention Study (WOSCOP) and Cholesterol and Recurrent Events (CARE) studies show that despite comparable serum cholesterol levels in the study groups, statin-treated patients had a significantly lower risk of CVD compared to age-matched placebo-controlled persons [63–65].

We are using statins as a therapeutic agent, or a *control variable* in the control-theoretic talk. We expect statins will control inflammation in the endothelial layer of the vascular artery slowing the inflammation process down, stabilising or even reversing it, to a limited extent

## Model simulations

The system of Eqs (1)–(3) constitutes a mathematical model capable of anticipating the size of atherosclerotic plaque in patients treated by statins. We notice that, unlike a similar model

proposed in [9] in that the terminal plaque size is modelled through a static equation (with 6 constants), the current model proposes a differential Eq (3) (with 5 constants) for the terminal plaque size. This equation is coupled with a differential equation for the plaque. Because of the coupling, the current model—arguably—better reflects the disease process than the other model. Also, the current model requires one constant less, which makes it more robust. Given nonlinearities and the coupling, system (1)–(3) will be solved numerically. The statin-treatment dependent IMT time profiles are shown in Fig 3.

In this figure, we can appreciate how statins may alter the growth process of the atherosclerotic plaque in heavy sufferers (on maintenance dialysis). The dotted (red) line that represents the plaque progression in an untreated patient, which tends to 1.678 by the patient's old age. The dashed lines show the effect of 80 [mg/d] of Atorvastatin may have on patients whose treatment start when they are, respectively, 40, 60 and 70 years old. In older patients, the IMT diminishes by up to 0.05 [mm] in the first 3-4 years and then resumes to grow but slower than without the treatment. The young patient's reaction to statins is monotonic and the IMT growth is slower that in the older patient. The effect of administering 40 mg/d Atorvastatin, shown by the solid (thin black) line, is similar but—expectantly—weaker.

We are pleased to notice that a reduction in the IMT growth as a result of statin therapy, so clearly visible in Fig 3, was reported in [66]. In their study (actually, a sub-study of the Japan Statin Treatment Against Recurrent Stroke—J-STARS), they looked into the impact of statins on IMT progression among 793 non-stroke Western patients with non-cardioembolic ischemic stroke. Patients were randomised into two groups: (1) receiving pravastatin (10 mg/d) and (2) not receiving any statins. At the beginning of the study, the IMT readings were not statistically different in these groups. However, after 5 years, the annual growth of IMT was significantly (statistically) reduced in the statin-treated group, compared to the control group. The authors of the trial concluded that the usual Japanese dose of pravastatin (10 mg/d) has significantly reduced the IMT progression among the non-cardioembolic stroke patients.

A figure, similar to Fig 3, was published in [9] using a slightly different model. The trajectories shown in Fig 3 have the same convergence points for $t = 90$ as those in [9]. However, the attentive reader will notice that in the dip, after statin administration has started, diminishes for younger patients. We believe that this may be so because, first, the plaque is thin, in absolute terms, in a young patient and there is not much thickness to be reduced by statins. Second, the *young* innermosts layer of the walls of large and medium-sized arteries is firm and less prone to deformation than *old*. We therefore believe that the current Fig 3 represents an improvement relative to the analogous figure in [9].

A general benefit of figures like Fig 3 is in visualising possible patient life-span IMT profiles, their sensitivity to statin doses and to patient age. In particular, this figure confirms substantial advantages of the early administration of statins.

We note that some authors e.g., [44, 67] consider the process of carotid thickening as linear. As a consequence, those authors use linear regression to identify this process. While parts of the curves in Fig 3 may look straight, they are solutions to nonlinear differential equations, which—possibly—better capture some of the physical processes responsible for the progression of vessels' thickening than straight lines.

## The viability kernels

### Disclaimer

The reasoning proposed in this section can be applied to any patient group characterised by the same atherosclerosis *condition*, see the time profiles in Fig 1; also see the dotted (red) line in Fig 3. That last time-profile corresponds to patients whose untreated atherosclerotic plaque

would reach 1.46 [mm] if they lived until 90 years. Let us call this high-risk CVD group G146. Of course, in real life, "similar" patients' profiles can lie above or below the aggregate profile. Furthermore, not all patients with the same condition, will react to statins in the same way. Therefore, caution needs to be used while interpreting the below quantitative conclusions.

## Viability kernels as "safety regions"

We call the pair [*age*, IMT] (where *age* is patient age) the atherosclerosis state. Consider the patients in G146 who are treated with statins. We would like to use our model to find the *viable* states, i.e., these states from which atherosclerosis can be slowed down. (The model will also tell the clinician which states are nonviable). This aim will be realised by framing the atherosclerosis process in *viability theory*—a mathematical theory of studying the evolution of dynamic systems defined in the state space, see e.g., [10] or [11].

The "viability kernel" is the key concept in viability theory. It encompasses the states from which the "system" can operate safely. In this context, a patient treated with statins constitutes the *system*, and safety means existence of statin therapies—i.e., controls—which lead the system's trajectory to a medically acceptable target whereas this trajectory remains within some prescribed bounds. (See [68] for a non-medical example of a viability problem with a target.) In our case, the viability kernel will be composed of the atherosclerosis states $(t, x)$ from which a statin therapy can keep the IMT below the lethal thickness level—$\underline{x}^*$—until the patient reaches an "old age"—$\underline{T}^*$.

A reason for us to use the viability kernel in analysing atherosclerosis is that the medical problem of controlling patient's IMT to a desired thickness can be modelled and solved by computing the viability kernel, for a mathematical model of atherosclerosis progression.

## Non-lethal atherosclerosis boundary

We contend that a patient from G146 whose non-treated IMT growth is characterised by the dotted (red) line in Fig 3 can live—*if* treated with statins—until $\underline{T}^* = 90$ [yo]. In that case their atherosclerotic plaque should remain under $\underline{x}^* = 1.15$ [mm].

We can see in Fig 3 that non-treated patients from G146 should undergo statin therapy otherwise their atherosclerotic plaque will grow to dangerous levels. In particular, the dotted (red) line suggest that the plaque will reach IMT = 1.15[mm], deemed fatal, when these non-treated patients are about 72 [yo]. This is easier to see in Fig 4 in which the critical level IMT = 1.15 [mm] is marked by the horizontal dash-dotted (blue) line. The dotted (red) line clearly crashes through the critical level 18-19 years before the "old" age.

Consider a patient from G146 whose statin treatment starts at $T = 70$ [yo]. If the patient is treated with the higher dose of statins ($s = 80$[mg/d]), then—as per Fig 3 top dashed line— their IMT can reach the target $x^* \leq \underline{x}^*$ in time $\mathcal{T} \geq \underline{T}^*$. However, on a smaller dose of statins ($s = 40$[mg/d]), the patient's IMT—see the top solid (black) line in both Figs 3 and 4—grows above $\underline{x}^*$ and they probably die before 90 [yo].

Because statins can have side effects, overdosing is not recommended. We will therefore propose which collections of states represent the safety regions for $s = 40$[mg/d] and, separately, for $s = 80$[mg/d]. A safety region for an intermediate dose can be obtained through interpolation.

## Statin therapy with $s = 40$[mg/d]

The vertical dash-dotted (blue) line in Fig 4 marks the patient "old" age i.e., $\underline{T}^* = 90$ on this line. As said above, we would like the patient to reach $\underline{T}^* = 90$ with IMT≤1.15[mm]. Starting

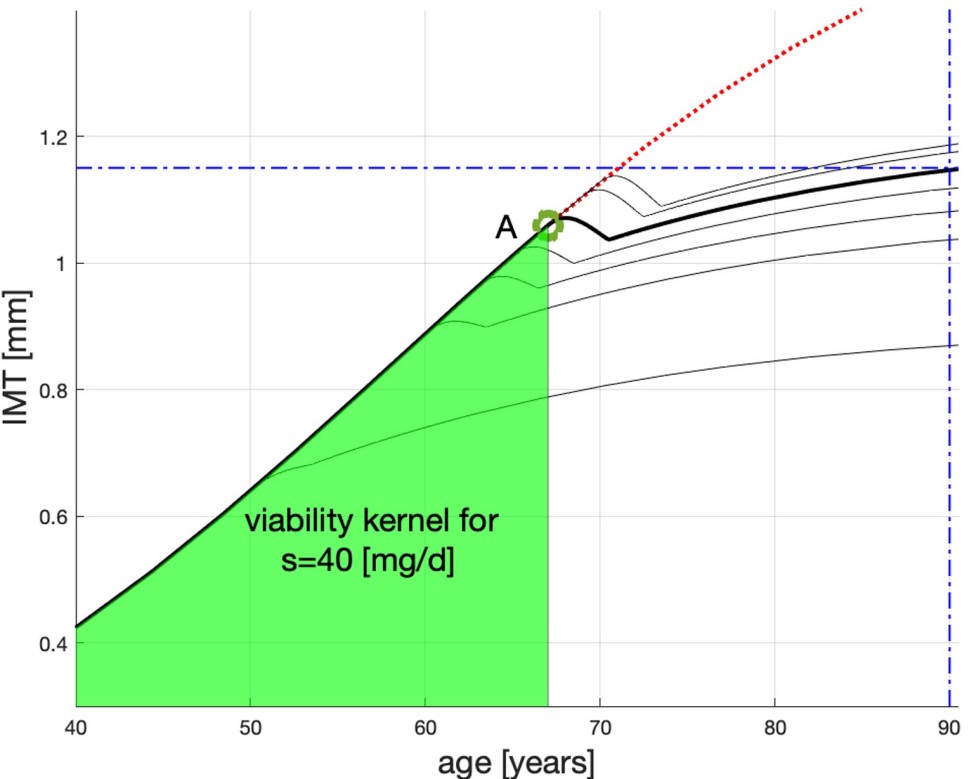

**Fig 4. Plausible age profiles for G146 patients treated with Atorvastatin 40 mg/d from: 50, 60, 63, 65, 67, 69, 70 yo.** The green area is the viability kernel for $s = 40$ mg/d.

the statin therapy with $s = 40$[mg/d] at $t = 70$[yo]—see the top thin solid (black) line in Fig 4 (same as the top thin solid line in Fig 3)—or $t = 69$[yo]—see the second from the top thin solid (black) line—does not guarantee the satisfactory outcome: both lines crash through $\underline{x}^* = 1.15$ [mm] before $\underline{T}^* = 90$.

The line that reaches the target—i.e., $\underline{T}^* = 90$ with IMT≤1.15[mm]—corresponds to the therapy which starts at $t = 67$[yo]. This is the solid thick (black) line which passes through point A (round green) (67, 1.059). The lines that originate below A represent the therapies which start at $t = 65, 63, 60$ and 50[yo]. All these therapy lines reach the target!

We will extend the above reasoning to G146 patients younger than 67 [yo] who have IMT below the disease progression line—solid (black) until A, dotted (red) after A. By interpolation, these patients treated with statins $s = 40$[mg/d] should have their IMT(90) $\leq \underline{x}^*$.

The area of atherosclerosis states from which the statin therapy $s = 40$[mg/d] controls a patient to the target is this therapy's *viability kernel*, coloured green in Fig 4.

## Statin therapy with $s = 80$[mg/d]

For G146 patients treated with $s = 80$[mg/d], the age $T = 70$ is the cutoff age above which no statin treatment can slow down the atherosclerosis process to the acceptable $x \leq \underline{x}^* = 1.15$ [mm] at $\underline{T}^* = 90$ [yo]. See the thick dashed (black) line passing through point B(70,1.128), marked as a square (beige), in Fig 5. For example, a 74 yo patient, with IMT(74) as on the dotted (red) line, is not expected to reach the target age $\underline{t}^* = 90$[yo] in good health i.e., with $x^* \leq 1.15$. This patient's therapy is represented by the top dashed (black) line in Fig 5. It crashes through $\underline{x}^* = 1.15$[mm] when the patient is 80[yo]. However, the patient whose therapy starts

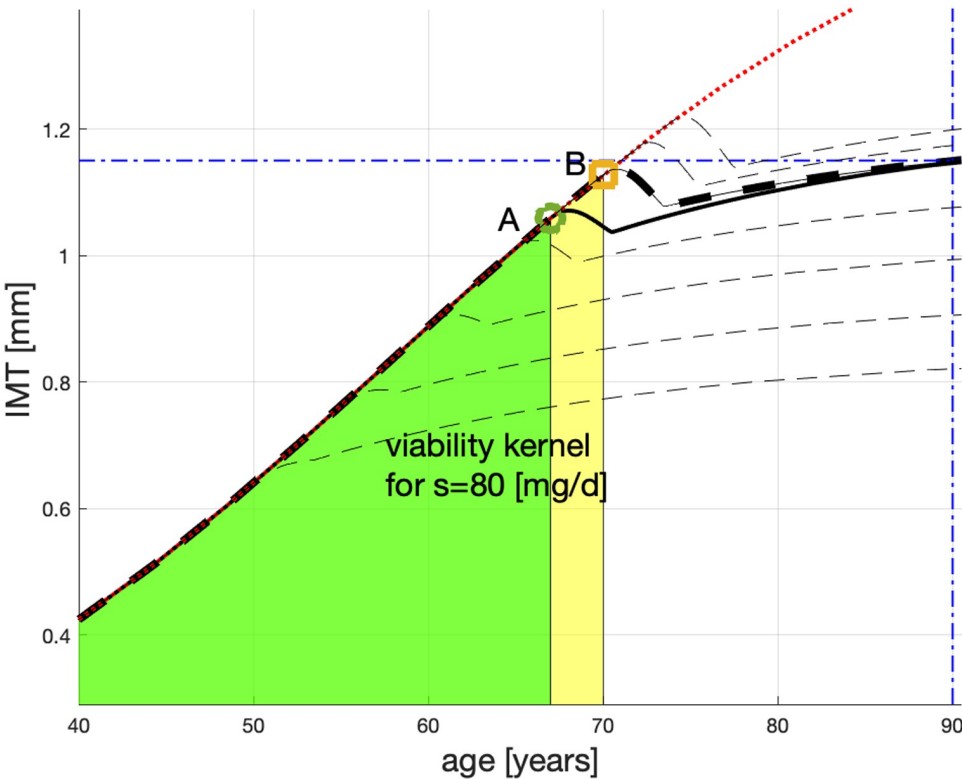

**Fig 5. Plausible age profiles for G146 patients treated with atorvastatin 80 mg/d from: 50, 60, 65, 70, 72, 74 yo.** The yellow area (overlapping the green area) is the viability kernel for $s = 80 \, \text{mg/d}$.

at point B, which is represented by the thick dashed (black) line, leads the patient to IMT(90) = 1.150[mm]. We therefore conclude that (70, IMT(70)) belongs to the *viability kernel*—i.e., the safety region, whereas (74, IMT(74)) does not. So do not the starting points for treatments that begin *after* $t = 70$[yo].

In Fig 5, we show the time profiles for statin therapies with $s = 80$[mg/d] as the dashed (black) lines. (The solid lines and the green area are as in Fig 4.) Those lines which start above B correspond to nonviable therapies; those below B—control the patient to a successful therapy conclusion and are therefore viable. By interpolation, the G146 patients younger than 70 [yo] whose IMT is below the disease-progression line should have their final IMT(90) $\leq \underline{x}^*$. We will call the area of atherosclerosis states from which the statin therapy with $s = 80$[mg/d] can control a patient to the favourable end the therapy's *viability kernel*. The corresponding viability kernel is shown as the yellow area (overlapping the green area) in Fig 5.

We note that $\dfrac{dc}{dt}$ is monotonic in $s$, see Eq (3), and so is its integral—$c(t)$. Therefore, for the statin doses—constant or variable—restricted to (40, 80), the corresponding atherosclerosis evolutions will remain between the evolutions emanating, respectively, from the points **A** and **B** in Fig 5.

## Concluding and methodological remarks

Atherosclerosis is a very complex disorder with many intermediate stages. A mathematical representation of this disease proposed in this paper should help understand its progression and suggest treatment. In particular, our model aims at assisting the doctors to quantify an

expected change of the plaque growth due to the statins administration. Furthermore, the *viability kernels*, and their complements, will suggest to the doctors when the statin administration is beneficial.

We are by-and-large satisfied that our graphs, obtained from the model, support the clinical facts established in the subject literature. While, at present, the model parameters have been calibrated, a clinical study that is being conducted will produce a large data base. This will allow us to estimate the model parameters and improve the model accuracy.

Our research consists of mathematical modelling of the dynamic process of atherosclerosis. In general, we follow the common approach of control theory to building and applying a dynamic process' model. This includes examining the underlying somatic process to determine which variables are key to this process. They are the *state* variables: IMT—the intima-media thickness, and *c*—patient wellbeing proxied by the plaque size at patient's death. The variable, the amount of which can be decided by the physician, and which has an impact on the state variables, is the dose of statins—the *control* variable. Next, mathematical formulae for the relationships between the studied variables are proposed and quantified. This can be done by estimating the model parameters using historical data. Should historical data be unavailable, the parameters may be calibrated using literature results. Finally, the estimated and/or calibrated formulae are used to carry out modelling experiments that might replace medical trials.

Given the control problem defined by the above state and control variables, together with their mathematical relationships, we have used viability theory to compute the viability kernels. These kernels define the patient conditions for which the statin application can succeed.

To the best of our knowledge, this paper is truly interdisciplinary and pioneers the use of viability theory in modelling dynamics and control of the atherosclerotic plaque.

## Appendix

### The cholesterol debate

Here we provide more arguments for why we have not included cholesterol in our model.

It has been found in the laboratory conditions that low-density lipoprotein cholesterol (LDL-C), responsible for transporting cholesterol to tissues, in its native form, does not induce foam cells formation, which are an important component of atherosclerotic plaque [69]. Also, relatively recent meta-analyses and systematic reviews, see [35, 70], have started to question the validity of the lipid hypothesis, as there is lack of an association between LDL-C and both all-cause and CVD mortality or this association is weak or even inverse.

If there was a positive statistically significant relationship between high LDL-C and CVD, then LDL-C of CVD patients would be statistically higher than among those without CVD. In many studies, see [71] involving 136,905 patients with AMI, almost half of them had admission LDL-C levels lower than normal. In another study [72], researchers revealed that lower LDL-C at admission was associated with *decreased* 3-year survival in patients with non-ST segment elevation myocardial infarction (NSTEMI). In an Austrian 2004 study [73] involving 67 413 men and 82 237 women (aged 20-95 years) followed for many years, high total cholesterol (T-C) was found to be weakly positively associated with coronary artery disease mortality. There was no association between T-C and mortality due to other CVDs, except that low T-C was found to be inversely associated with CVD mortality in women over 60 years of age. Moreover, it is not completely clear whether the maxim *the lower, the better cholesterol* for CVD and ischemic heart disease (IHD) can be applied to general populations with a low risk of heart disease mortality, see [74]. These researchers prospectively followed up 503 340 Koreans aged 40-79 (at the beginning of the study), who participated in routine health checkups via linkage to national mortality records. The assessment was done two times, at the beginning and after 10

years. Surprisingly, they found nonlinear associations with T-C, i.e., U-curves for overall CVD, and reverse L-curve for IHD. T-C high levels of 180-200 `mg/dL` were associated with the lowest CVD mortality. Here we should remember that cholesterol is an essential biomolecule for the proper functioning of all human cells, and it should be not surprising that there are several consequences due to the aggressive lowering of cholesterol in the body [75].

These findings have led to the "re-discovery" of chronic inflammation, a physiological response of the innate immune system that maintains a constant internal environment, while being exposed to ever-changing environmental factors, regardless of their cause. The inflammatory response is aimed at reducing the factor to restore tissue homeostasis. In this situation, an approach in which elevated cholesterol is not the causative agent or underlying biochemical mechanism responsible for endothelial dysfunction and atherosclerosis appears to be justified.

The innate immune system treats the accumulation of excess LDL-C in plasma as an adverse event. Therefore, an inflammatory response in the endothelial wall is promoted to reduce the risk by removing excess LDL and oxidised LDL-C (Ox-LDL) from the bloodstream to the endothelium, where they are absorbed by migrating monocytes for final removal. So it is not cholesterol itself, but the inflammatory reaction as a response to its accumulation, which modifies it together with oxidative stress.

## Acknowledgments

We thank two referees and the Academic Editor for their queries and suggestions that helped us to improve the paper.

## Author Contributions

**Conceptualization:** Dorota Formanowicz, Jacek B. Krawczyk.

**Data curation:** Dorota Formanowicz, Jacek B. Krawczyk.

**Formal analysis:** Dorota Formanowicz, Jacek B. Krawczyk.

**Funding acquisition:** Dorota Formanowicz.

**Investigation:** Dorota Formanowicz, Jacek B. Krawczyk.

**Methodology:** Dorota Formanowicz, Jacek B. Krawczyk.

**Resources:** Dorota Formanowicz, Jacek B. Krawczyk.

**Software:** Jacek B. Krawczyk.

**Validation:** Dorota Formanowicz, Jacek B. Krawczyk.

**Visualization:** Jacek B. Krawczyk.

**Writing – original draft:** Dorota Formanowicz, Jacek B. Krawczyk.

**Writing – review & editing:** Dorota Formanowicz, Jacek B. Krawczyk.

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
