## [Decision Letter · Decision Letter 0]

24 Jul 2020

PONE-D-20-18089

Controlling the thickness of the atherosclerotic plaque by statin medication

PLOS ONE

Dear Dr. Formanowicz,

Thank you for submitting your manuscript to PLOS ONE. After careful consideration, we feel that it has merit but does not fully meet PLOS ONE’s publication criteria as it currently stands. Therefore, we invite you to submit a revised version of the manuscript that addresses the points raised during the review process. Particularly, the comments regarding the expliciy formulation of vaiability kernels.

We look forward to receiving your revised manuscript.

Kind regards,

Míriam R. García

Academic Editor

PLOS ONE

Journal Requirements:

2. During our internal evaluation of the manuscript, we found significant text overlap between your submission and the following previously published works:

https://www.mdpi.com/1422-0067/20/3/785/html

Please revise the manuscript to rephrase the duplicated text, cite your sources, and provide details as to how the current manuscript advances on previous work. Please note that further consideration is dependent on the submission of a manuscript that addresses these concerns about the overlap in text with published work.

3. Please clarify in your Data availability statement whether any existing datasets were used, and whether the code for the model has been made available.

Reviewers' comments:

Reviewer's Responses to Questions

**Comments to the Author**

1. Is the manuscript technically sound, and do the data support the conclusions?

Reviewer #1: Yes

Reviewer #2: Yes

2. Has the statistical analysis been performed appropriately and rigorously? 

Reviewer #1: Yes

Reviewer #2: N/A

3. Have the authors made all data underlying the findings in their manuscript fully available?

Reviewer #1: Yes

Reviewer #2: Yes

4. Is the manuscript presented in an intelligible fashion and written in standard English?

Reviewer #1: Yes

Reviewer #2: Yes

5. Review Comments to the Author

Reviewer #1: This paper presents a model for control-theoretic with the viability kernels. The authors claim that this modelling the atherosclerosis progression is original. But I do not understand how the viability kernels apply to their model. I would like to suggest writing the explicit formulation of viability kernels in the manuscript. I have the following comments to better understand the mathematical model

1) In eq. 3, how can you guarantee the continuity at t=T?

2) What's the dynamics of s(t)? it looks like you choose a constant in your simulation...

3) How the viability kernels contribute to Eq. (3) and Eq. (1)?

Reviewer #2: The paper is a significant step in modeling atherosclerosis. The approach used by the authors is very interesting interns of modeling and for clinical applications later on. I do not have additional comments.

The paper is well written and I recommend its publication in Plos One.

6. PLOS authors have the option to publish the peer review history of their article (what does this mean?). If published, this will include your full peer review and any attached files.

Reviewer #1: No

Reviewer #2: No

---

## [Author Response · Author response to Decision Letter 0]

13 Aug 2020

See Response to Reviewers.pdf uploaded to 'File Inventory'.

---

## [Decision Letter · Decision Letter 1]

16 Sep 2020

Controlling the thickness of the atherosclerotic plaque by statin medication

PONE-D-20-18089R1

Dear Dr. Formanowicz,

We’re pleased to inform you that your manuscript has been judged scientifically suitable for publication and will be formally accepted for publication once it meets all outstanding technical requirements.

Kind regards,

Míriam R. García

Academic Editor

PLOS ONE

Additional Editor Comments (optional):

Reviewers' comments:

Reviewer's Responses to Questions

**Comments to the Author**

1. If the authors have adequately addressed your comments raised in a previous round of review and you feel that this manuscript is now acceptable for publication, you may indicate that here to bypass the “Comments to the Author” section, enter your conflict of interest statement in the “Confidential to Editor” section, and submit your "Accept" recommendation.

Reviewer #1: All comments have been addressed

2. Is the manuscript technically sound, and do the data support the conclusions?

Reviewer #1: Yes

3. Has the statistical analysis been performed appropriately and rigorously? 

Reviewer #1: N/A

4. Have the authors made all data underlying the findings in their manuscript fully available?

Reviewer #1: Yes

5. Is the manuscript presented in an intelligible fashion and written in standard English?

Reviewer #1: Yes

6. Review Comments to the Author

Reviewer #1: (No Response)

7. PLOS authors have the option to publish the peer review history of their article (what does this mean?). If published, this will include your full peer review and any attached files.

Reviewer #1: No

---

## [Editor Report · Acceptance letter]

24 Sep 2020

PONE-D-20-18089R1 

Controlling the thickness of the atherosclerotic plaque by statin medication 

Dear Dr. Formanowicz:

I'm pleased to inform you that your manuscript has been deemed suitable for publication in PLOS ONE. Congratulations! Your manuscript is now with our production department. 

Kind regards, 

on behalf of

Dr. Míriam R. García 

Academic Editor

PLOS ONE